# Moisture as a Driver of Long-Term Threats to Timber Heritage—Part I: Changing Heritage Climatology

**Jenny Richards [1,2] and Peter Brimblecombe [3,*]**

1   St John's College, Oxford University, Oxford OX1 3JP, UK
2   School of Geography and the Environment, Oxford University, Oxford OX1 3QY, UK
3   Department of Marine Environment and Engineering, National Sun Yat-Sen University, Kaohsiung 80424, Taiwan
*   Correspondence: p.brimblecombe@uea.ac.uk

**Abstract:** Timber is widely used in the construction of buildings on a global scale, but it is sensitive to degradation. Moisture notably poses a risk to timber decay, and this is likely to change significantly during the 21st century if a high emission scenario occurs. Global HadGEM3 model output was used to map projected changes in relative humidity range, seasonality of relative humidity, time of wetness, wind-driven rain, salt transitions and potential for fungal attack (Scheffer Index). In the Congo Basin, Great Plains (USA) and Scandinavia, humidity ranges are likely to increase along with seasonal change. In many parts of the tropics, time of wetness is likely to decrease by the end of the century. Increases in days of wind-driven rain are projected for western Russia, eastern Europe, Alaska, western Canada and Southern Brazil and Paraguay. Drylands have historically had a low salt risk, but this is projected to increase. In the future, a broad extension of fungal risk along the Himalayas and into central China seems likely, driven as much by temperature as rainfall. The picture presented suggests a slightly less humid heritage climate, which will redistribute the risks to heritage. Mapping global pressures of timber decay could help policymakers and practitioners identify geographically disparate regions that face similar pressures.

**Keywords:** deterioration; global climate change; cultural heritage; built heritage; wood; HadGEM3; heritage management

## 1. Introduction

Timber has been widely used since prehistoric times in both major buildings and more humble dwellings. It has also been present in other forms of heritage from structures, such as bridges, through to movable objects, such as furniture, sculpture and musical instruments. Given its wide application, the process of timber deterioration has long been of concern. The pressures driving deterioration may increase under a changing climate as expressed in the ICOMOS *Principles for the Conservation of Wooden Built Heritage* as a need to "recognize the vulnerability of structures made wholly or partially of wood in varying environmental and climatic conditions, caused by (among other things) temperature and humidity fluctuations, light, fungal and insect attacks, wear and tear, fire, earthquakes or other natural disasters, and destructive actions by humans".

Water and moisture relations are especially important for wood. These can drive physical changes and mediate biological and chemical processes that cause deterioration in timber [1,2]. To assess the impact of climate on historic timber, traditional meteorological and climatological parameters need to be tuned and refined to reflect potential threats as a *heritage climate*. Heritage climatology is an expression of meteorological characteristics that affect tangible heritage [3–5]. The projected change to global climate over the 21st century poses a complex challenge to the management of heritage, with the nature of change uncertain in many regions. Previous work on the impact of climate on heritage has

been predominantly studied at site [6], country [7,8] and regional [3,9] scales and less so on the global level. Nevertheless, a global scale gives a context that can inform management of a specific site or policy for a region.

### 1.1. Distribution

Timber tends to dominate as a construction material in areas where (i) suitable timber is available, (ii) it is an economical resource, and (iii) it is socially and architecturally desirable. As examples, in Northern Europe, abundant forests have led to a long tradition of wooden construction [10], and timber is an important material in the vernacular architecture of Africa [11]. Additionally, wood has found a prominent place in major buildings of China and Japan [12].

The wide geographical distribution of timber heritage means that it interfaces with a wide range of hygrometric conditions from drylands to rainforests. Given the sensitivity of timber heritage to moisture, long-term climate change could exacerbate deterioration processes in timber. This threat has been particularly noted in areas where the frequency of extreme rainfall and winds, along with the potential for warmer and possibly damper climates, occurs [13–15]. Similarly, the effect of climate change on insects that burrow into timber has been studied, with insects being more abundant and able to complete more reproduction cycles under future climates [16,17].

This paper focuses on timber heritage and the impact of rainfall, temperature and relative humidity to explore the importance of changing heritage climates.

### 1.2. Threats

Moisture is an important control on the physical dimensions of wood, with fluctuations in moisture content causing swelling and shrinking of the material. Persistent variations in relative humidity and moisture interacting on the carved wood surfaces (e.g., in sculpture) can cause such objects to weaken and crack.

Rot is another common process threatening timber heritage. Fungal attack is mediated by climate factors: temperature, water or exposure to high humidity [1,18]. The changes are often the result of fungi, such as *Monilinia fructicola*, *M. laxa*, *Serpula lacrymans*, *Gloeophyllum trabeum* and *Coniophora puteana*, and are commonly described as brown rot (dry rot) [19], white rot or soft rot [20]. Brown rot fungi are often from tropical climates and southern temperate zones [21]. White rot fungi are active over the temperature range 18–32 °C, but may be active up to 45 °C. [22].

Timber is also threatened by insects [23,24], such as carpenter ants (*Camponotus* spp.), termites (Epifamily: Termitoidea), bark beetles (subfamily Scolytinae), longhorn beetles (Family: Cerambycidae), weevils (Superfamily: Curculionoidea), and powderpost beetles (superfamily Bostrichoidea). The deathwatch beetle (*Xestobium rufovillosum*) is especially well-known in historic structures as it prefers aged oak timber rather than softwood. Wood is generally susceptible to insect attack at high humidity. Under humid, damp conditions, timber can be softened through fungal decay, meaning that insect larvae can more easily tunnel into the wood using the cellulose and hemicellulose as a food [25].

As with many other heritage building materials (e.g., stone, brick and earth), the deterioration of timber can also be caused by salts [26,27]. For example, salt deterioration in timber has been described from polar regions in explorer huts [27]. Salts can also arise from groundwater or due to the activities occurring within the buildings (e.g., fish curing). One of the most common salts is sodium chloride, which dissolves or crystallises at 75.5% relative humidity and is derived from sea spray and road salts [26], but in buildings located inland, nitrates and urea contribute to the salts present. This state change can cause physical stresses within the timber, and in cases of high salt concentrations, defibration can also occur [27]. The severity of salt deterioration processes can be estimated from environmental conditions, typically using changes in relative humidity.

Other mechanisms of damage, such as abrasion, wear and tear, fire, earthquakes or other natural disasters, are not discussed here due to our focus on moisture, although they

can be readily explored as for example at the temples and shrines of Nikkō and expedition huts on Antarctica [28,29]. In addition, we focus on outdoor timber, so we do not discuss indoor heritage [9,30] or waterlogged archaeological wood [31,32].

*1.3. Approach*

This study investigates the moisture-related pressures imposed by climate on timber heritage, here taken primarily as outdoor buildings. These pressures can represent threats to conservation by driving processes of deterioration. We used global climate models to identify regions exposed to high moisture-related threats and to examine the changes over time and space to reveal regions of particular sensitivity.

We used maps as the primary tool for assessing the results from the climate model. Maps are widely used in the heritage sciences as a method to convey spatial data [33–36]. Although the transfer of the data presented in maps into practice is nontrivial [14], they are useful for gaining a broad-scale understanding of the effect of change on heritage across the world.

Global maps also provide a useful tool to represent projections of future conditions and give an understanding of processes and impacts that are not constrained to a single country or region. In a world where so much heritage research has focussed on mid-latitude, high-income countries, these global scale outputs are important [37,38]. This scale of approach facilitates assessment and comparison of threats from across multiple regions, informing decision making at a strategic level. A high-level understanding of threats is important for agenda setting within regional- and national-level institutions. The importance of addressing these risks at a local scale is also recognised and is discussed in a later publication [39]. In particular, in this study, we aim to globally assess the risk posed by past and future moisture-related processes, which drive deterioration of historic timber.

## 2. Methods

This study used global climate data available from historic, contemporary and future projections derived from the HadGEM3-GC31-MM model within the CMIP6 ensemble [40]. We used four 30-year time periods: 1850–1879 (historic; to set a baseline); 1984–2013 (recent past; many observational datasets available); 2025–2054 (near future; current planning period) and 2070–2099 (far future; sense of overall direction), with future projections determined using the Shared Socio-economic Pathways 585, a scenario based on a high emission future (i.e., a worst-case scenario). We used mean daily relative humidity, temperature, surface wind speed and precipitation (all at 2 m above ground level) from the climate model. These outputs were converted to heritage climate variables, relevant to external pressures on timber. The use of daily timesteps, rather than using higher temporal resolution, captures the response time of historic timber to moisture changes studied in this paper.

Six heritage climate parameters were calculated as moisture-related drivers of timber deterioration: (i) annual relative humidity range ($\Delta RH_a$); (ii) seasonality of relative humidity, (iii) time of wetness, ToW; (iv) wind-driven rain, WDR; (v) salt transitions; and the (vi) Scheffer Index for fungal risk (*Sch*). No distinction was made between untreated timber and that with pesticides or surface coatings, although these would respond more slowly to climate and biological threats; they are looked at in more detail in Brimblecombe and Richards [39].

- **Relative humidity range** was calculated as the annual range in mean monthly relative humidity (%), where the annual range ($\Delta RH_a$) is $RH_{max} - RH_{min}$; $RH_{max}$ is the RH of the month with the highest mean RH in a given year, and $RH_{min}$ is the minimum mean monthly RH in the same year. These differences were summed and divided by 30 to give the average range over the 30-year period. It should be noted that this value is not the same as calculating the difference in the highest and lowest RH values from an averaged 30-year dataset, which are commonly presented in climate summaries. The range expressed in such climatological norms does not reflect the humidity stress experienced

by wood each year. Assessing the difference between the highest and lowest month per year (rather than over a 30-year timeframe) captures greater variations in humidity conditions, so the ranges are notably larger than represented by 30-year climatologies.

- **Seasonality of relative humidity** identified the month in a given year that had the highest and lowest mean RH. The modal month for each 30-year time period characterised the seasonality.
- **Time of wetness** is often determined as the number of days per year when relative humidity is >80%, and temperature is >0 °C [41]. This was adopted here, and we also note that a relative humidity of 80% is part of standard methods for laboratory evaluation of insect (e.g., termite) damage to and consumption of wood [42].
- **Wind-driven rain** is defined as the number of days per year when rain is >4 mm; mean wind speed is >2 m s$^{-1}$, and temperature is >0 °C; adapted from Rydock et al. [2].
- **Salt transitions** were expressed as the number of cycles per year where the mean daily RH crossed 75.5%, to account for sodium chloride crystallisation. A daily transition was adopted to account for change in the phase of salts at the surface as in Grossi et al. [43].
- **Scheffer Index** estimates the risk of fungal attack expressed in the equation: $Sch = \Sigma (T_m - 2)(D - 3)/16.7$, which represents the sum over twelve months for the monthly mean temperature ($T_m$) and number of days in the month with $\geq$0.3 mm of rain [44,45]. This index has been frequently used at specific locations (e.g., Japan [29], Korea [7,15], Norway [8], Switzerland [6], the UK [46] and the USA [47]).

The heritage parameters were calculated for each 30-year period and averaged to provide a mean value. In the case of WDR and the Scheffer Index, which use multiple climate variables, these parameters were also run with only one climate variable active to assess the contribution of individual parameters. Box and whisker plots were used to present the RH range data. The box is bounded by the 25th and 75th percentiles, with the median denoted by the central line in the box. The whiskers represent the range of all other points, except those that are deemed as outliers. An outlier is considered to be any value that lies over 1.5 times the interquartile range below and above the 25th and 75th percentiles. We used the Mann–Whitney test, rather than the *t*-test, to assess significant differences in results between time periods, because of the nonparametric nature of our data.

## 3. Results

Our results show how the threat posed by heritage climate to timber heritage differs over space and time. These contribute to a strategic understanding of the past, present and future risks posed by moisture parameters to timber heritage across the world. The results should be interpreted in the context of broad groups of timber heritage within given regions, as microclimatic variations will influence the extent of risk posed by the heritage parameter at scale of sites, buildings or objects [6].

### 3.1. Relative Humidity Range

Figure 1 shows the global change in the RH range across the four time periods: the 19th century (1850–1879); recent past (1984–2013); the near future (2025–2054); and the far future (2070–2099). In general, the lowest RH ranges are found in polar and equatorial regions and areas with subtropical high pressure (e.g., the Sahara). In the 19th century and recent past, a low annual variability in the Amazon and Congo Basins and in Southeast Asia is particularly notable. Changes can be subtle, especially in the past, so we provide animated versions of the global maps in the supplement. An increase in the RH range is projected (Figure 1d) for the Great Plains of North America (extending into Northern Canada) and Southeast China, and at higher latitudes, this is also apparent across Western Europe and Scandinavia. This is likely to affect the timber heritage in these regions, which includes vernacular wooden structures in North America, ancient wooden temples and pagodas of China and Europe's timber-framed buildings, stave churches and carved wooden statues.

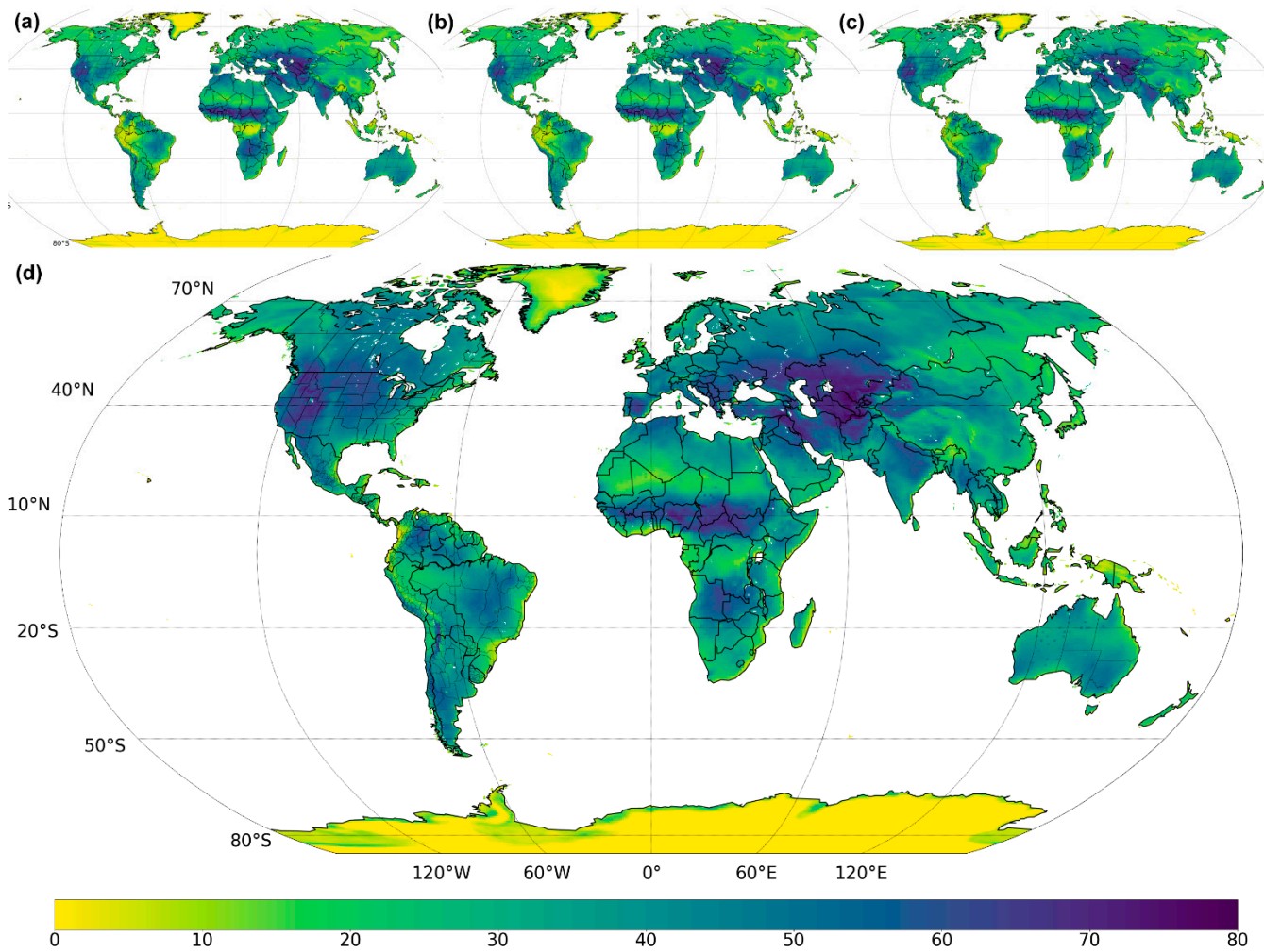

**Figure 1.** Mean annual relative humidity range ($\Delta RH_a$) for the periods (**a**) 1850–1879, (**b**) 1984–2013, (**c**) 2025–2054 and (**d**) 2070–2099. See also Video S1 (as an animated GIF) in Supplementary Materials.

The RH of the Congo is high throughout the year, but by the end of the 21st century, the RH range in the tropics is projected to increase substantially, a result of less humid conditions becoming more common in a warmer climate (Figure 2a). In the Congo, all months are projected to be less humid by the end of the 21st century, which is particularly apparent between January and March (Figure 2a).

Other mid- and high-latitude locations show similar trends in the RH range to that seen in the Congo, with increases throughout the 21st century (Figure 1). The Great Plains region of the USA, here as Wisconsin, Iowa and Minnesota, shows a notable decrease in summer RH for the far future (Figure 2b), and Scandinavia shows an increasing range, but winters remain damp (Figure 2c). The increase in the RH range in these mid- to high-latitude regions is predominantly caused by lower humidity in summer months, with some winter months also showing slightly higher humidity. In the Congo, the month with the maximum RH has shifted to later in the year; from 1850 to 2054, it predominantly occurred in April or May, but in the last 30 years of the 21st century, it is projected to be in July. The Mann–Whitney test suggests these periods are significantly different ($n_a = 90$; $n_b = 30$; $p_2 = 0.0271$). Typically, it is temperature that is seen as an important control on insect growth [16]. However, under a drier future in tropical areas, it is likely that a humidity decrease could be a more important factor than temperature in the declining diversity of termites in tropical forests [48]. In these regions, the prevalence of drier conditions

means less moisture is likely to be available for deterioration processes, particularly those biologically mediated ones.

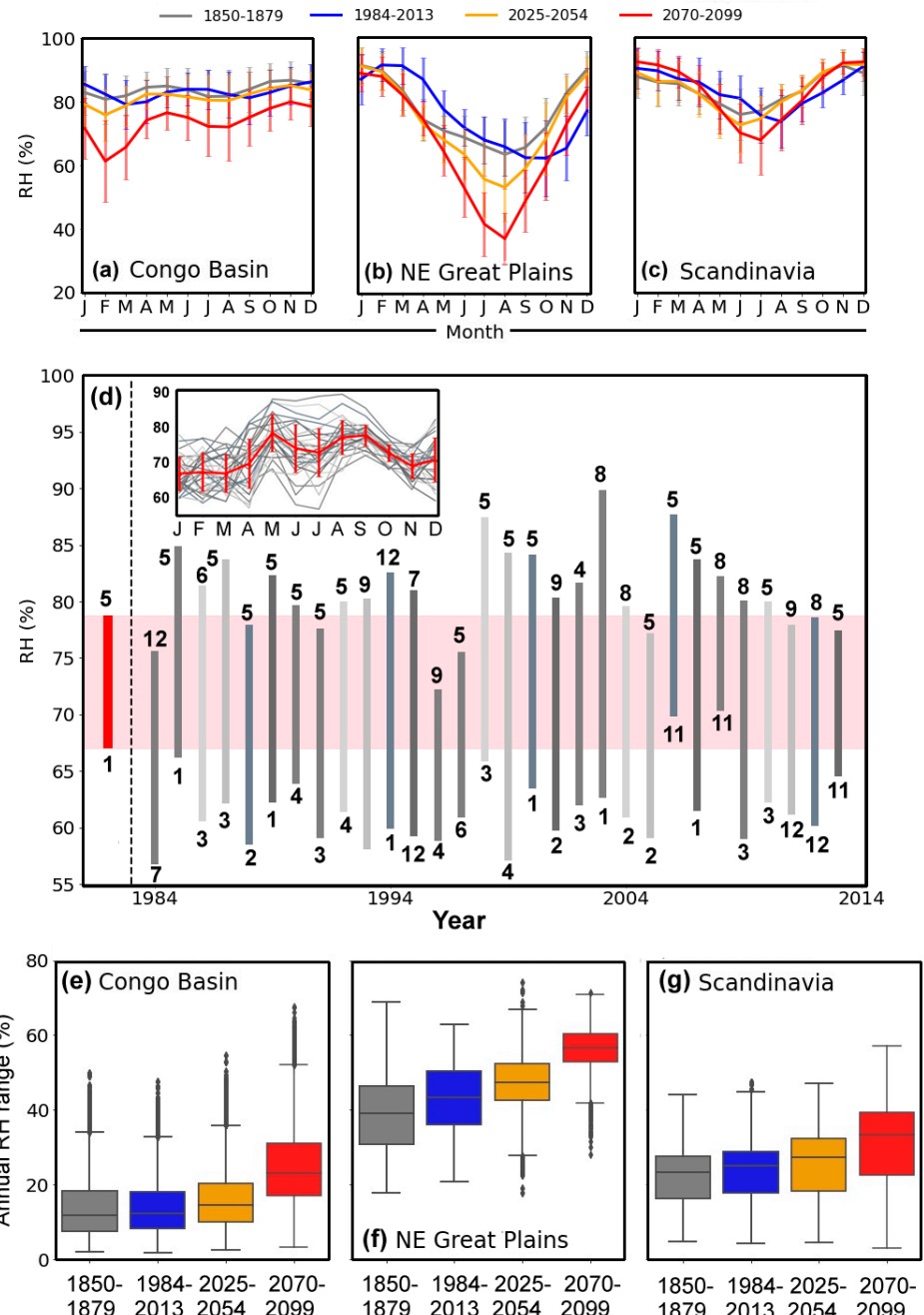

**Figure 2.** (**a**–**c**) The 30-year mean monthly relative humidity (RH%) for the time periods 1850–1879 (grey), 1984–2013 (blue), 2025–2054 (yellow) and 2070–2099 (red) for (**a**) the Congo Basin, (**b**) the Northeast Great Plains region of the USA, here as Wisconsin, Iowa and Minnesota and (**c**) Scandinavia (Norway, Sweden and Finland). Error bars show one standard deviation from the mean. (**d**) The range of RH between 1984 and 2013 for a single grid cell from the Congo Basin. The range of the 30-year mean monthly RH (red) is compared with each of the annual RH ranges (grey) for each of the 30 individual years. The months with the highest and lowest RH are denoted by the two numbers at the bottom and top of the bars. The inset displays the annual variation for each of the 30 years, and the red line displays the mean monthly RH and standard deviation for the period. (**e**–**g**) Box and whisker plots of the 30-year mean relative humidity range for the four time periods in (**e**) the Congo Basin, (**f**) the Northeast Great Plains and (**g**) Scandinavia.

It is important to consider that when the range is calculated from the average monthly humidity across a 30-year period, the range can seem quite small compared to when the range is calculated for each individual year in a 30-year period and then averaged. In this cell, the shorter red bar shows that the range in averaged monthly RH over a 30-year period is 11.7% (i.e., May (5) maximum 78.7% and January (1) minimum of 67%). The averaged range in RH of each of the 30 years is larger (19.4 ± 3.5%), shown in Figure 2d by the longer grey bars.

*3.2. Seasonality of RH*

The month most frequently associated with maximum RH (modal month) in the near past and the far future is shown in Figure 3. Globally, there is often a shift to the most humid months occurring earlier in the year. Large changes in seasonality are present in North and South America, with the most humid month in the USA moving from February to March to earlier in the winter, i.e., December–January, while in Central Brazil, there is a notable shift from the maximum RH occurring in April–May to February–March. In Western Australia, the season moves from August to September in the recent past back to June–July by the end of the 21st century. Smaller shifts in the most humid month are found in many other locations. For example, from February–March to December–January across much of Europe and Northern and Southern Africa and from September–October to August–September in Central and Western Africa. For India, Nepal and the Tibetan region, the most humid month in the recent past has been September, but by the end of the century, this is projected to occur in August. While shifts of a month or so seem small, these could result in noticeable changes to the timing of growing cycles. It may also be important to the cycle of human and cultural activities [29,49] or to heritage managers in determining when would be the most effective point in the year to implement conservation and management strategies.

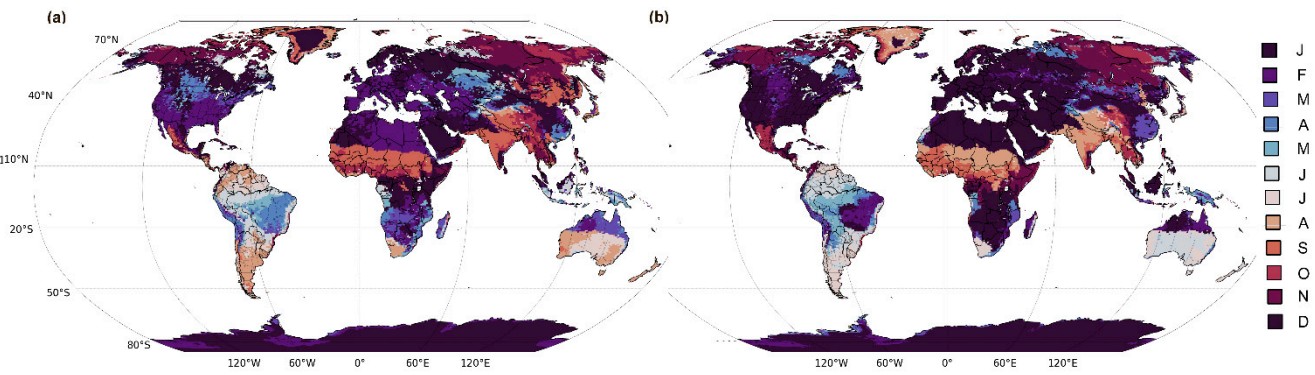

**Figure 3.** Modal month with the maximum monthly RH (**a**) 1984–2013 and (**b**) 2070–2099. See Supplementary Materials, Figure S1 and Video S1 for all four time periods. See also Video S2 (as an animated GIF) in Supplementary Materials.

Changes in seasonality in the modal month with minimum monthly RH were also found, but the changes were less notable. The results can be found in Supplementary Materials, Figure S2 and Video S3.

*3.3. Time of Wetness*

Figure 4 shows the ToW across the globe from the 19th century through to the end of the 21st century. The changes and global geographical distribution of pressures in the future for ToW is similar to that of the RH variation, with notable increases in the upper reaches of the Amazon, the Congo Basin and Southeast Asia. These regions have had lengthy periods of wetness in the past (>300 days per year), but the areas where this is persistent are projected to shrink in the near future (2025–2054), with substantial declines seen by the end of the century (Figure 4d). In cities and smaller settlements in these regions,

e.g., Tefé, a municipality in Amazonas in northern Brazil or in the Congo, Kisangani (formerly Stanleyville), the capital of Tshopo province, where timber has been used in both vernacular and old colonial buildings, such decreases in time of wetness could reduce the risk of insect damage as timber will have shorter periods of high moisture content.

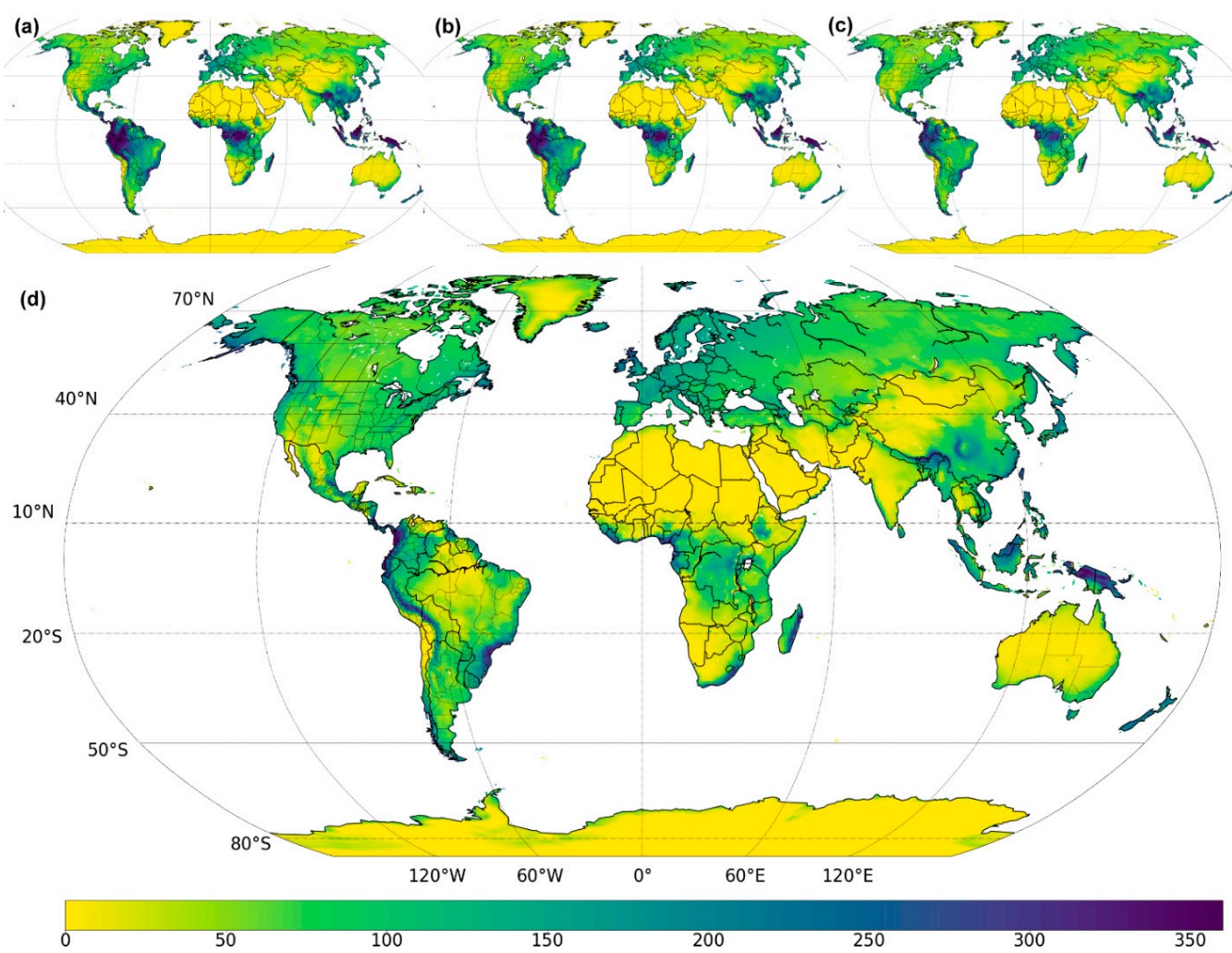

**Figure 4.** Time of wetness (days per year) for the periods (**a**) 1850–1879, (**b**) 1984–2013, (**c**) 2025–2054 and (**d**) 2070–2099. See also Video S4 (as an animated GIF) in Supplementary Materials.

The length of the wetness period in the high-altitude areas of the New Guinea Highlands and the Tibetan Plateau region are projected to remain high over the 21st century (Figure 4). This continued pressure from lengthy periods of wetness means that the risk of insect attack on historic timber in these regions remains high.

Perhaps of greater concern are increases in future ToW in regions where warming means more days above freezing, e.g., Russia, Canada and Antarctica (Figure 4). For example, in coastal areas of Antarctica, projections suggest that rising temperatures could lead to a more than three-fold increase in days of wetness by the end of the 21st century. Antarctic wooden heritage, such as the buildings associated with polar exploration [27] or industry [50], would be exposed to new drivers of deterioration over the next century. As timber heritage in these regions is already exposed to deteriorative conditions [28], the addition of further pressures from ToW could have a substantial impact on future deterioration.

### 3.4. Wind-Driven Rain

Figure 5 shows the pressure on timber heritage caused by WDR across the four time periods used in this study. Globally, changes in future pressures from WDR are low in dryland regions, but most frequent in the tropics, mid-latitudes and coastal regions with strong prevailing winds (e.g., eastern North America, south-eastern South America, northwestern Europe, West Africa and coastal regions of Southeast Asia). Many regions are projected to face increasing days of WDR by the end of the 21st century (e.g., Western Russia, Eastern Europe, Alaska and Western Canada). There are also increases in southern Brazil and Paraguay, where the style of timber architecture heritage has been influenced by German and Italian immigration. Additionally, increasing WDR will be expected in an area stretching from Sierra Leone and Liberia, across Central Africa to Sudan. In West Africa, in particular Freetown, Sierra Leone, it may affect North American-style timber houses found as a result of immigration after the abolition of slavery in the 19th century. As many heritage sites are found near the coast, this overlap of geographic distribution of pressures and location of heritage increases challenges for the conservation of timber heritage.

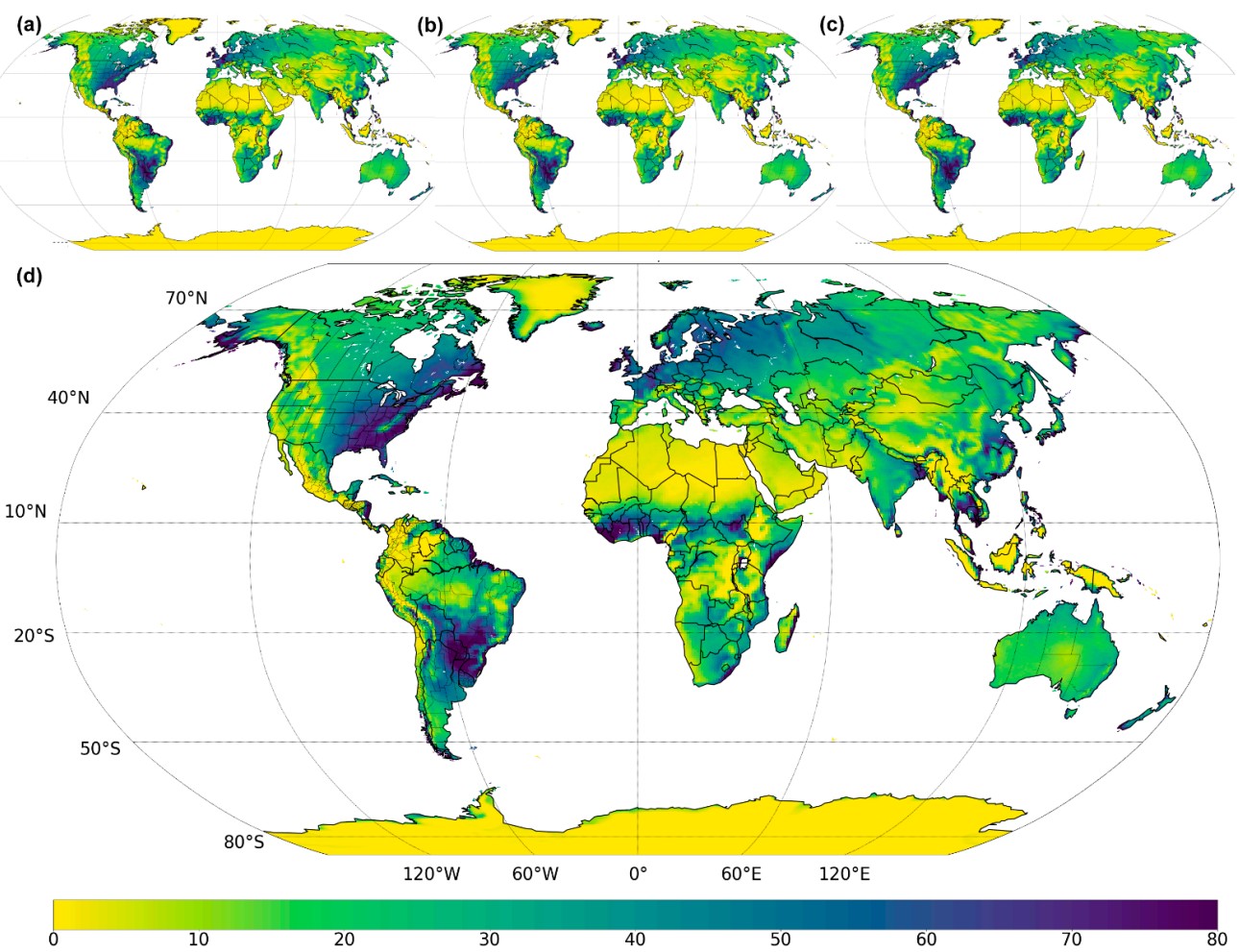

**Figure 5.** Average number of days of wind-driven rain each year for the periods (**a**) 1850–1879, (**b**) 1984–2013, (**c**) 2025–2054 and (**d**) 2070–2099. See also Video S5 (as an animated GIF) in Supplementary Materials.

Future change in pressures imposed by WDR are predominantly driven by changes in rainfall rather than wind speed (Figure 6). As climate models, such as HadGEM3, can underestimate rainfall in regions, such as parts of Southern Africa [51,52], the risk posed by WDR in these areas may be uncertain.

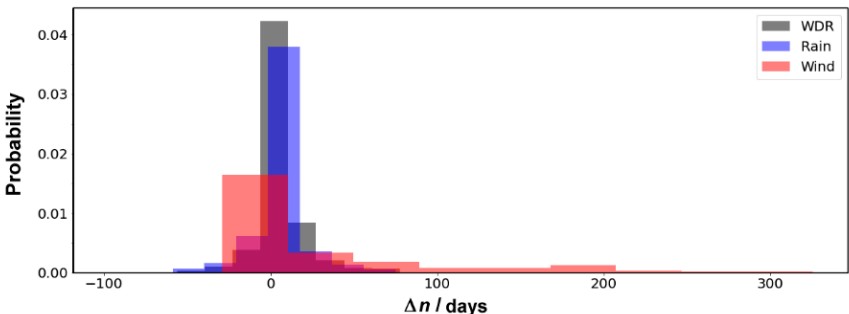

**Figure 6.** The probability of the difference in the number of days ($\Delta n$) of (i) wind-driven rain (WDR), (ii) days with rain >4 mm d$^{-1}$ and (iii) days with wind >2 m s$^{-1}$ between the far future (2070–2099) and the near past (1984–2013) across all land masses.

### 3.5. Salt Transitions

The modelled number of salt transitions for sodium chloride, occurring in timber heritage between the 19th and 21st centuries is shown in Figure 7. Regions experiencing frequent salt transitions of more than 30 per year are found across the globe (e.g., North America, Europe, China, Japan, New Zealand and southern South America and coastal Antarctica, Greenland, Australia and South Africa).

**Figure 7.** Salt transitions per year for the periods (**a**) 1850–1879, (**b**) 1984–2013, (**c**) 2025–2054 and (**d**) 2070–2099. See also Video S6 (as an animated GIF) in Supplementary Materials.

In terms of change, Figure 7 shows that some dryland regions that historically have had few salt transitions (e.g., Central Australia, the Middle East and the Sahara) will in future show increases in the projected pressure as they become more humid. By the end of the century, large increases are projected for western South America, Central Africa and Southeast Asia. These regions also experience an increased RH range (Figure 2), largely because the drier months will have lower RH in the future, which drives an increase in the number of transitions. Some small decreases are also projected in the eastern USA, Mediterranean and Europe as noted by Grossi and Brimblecombe [43].

### 3.6. Scheffer Index

The change in Scheffer Index for the four periods under investigation was small in comparison to the range of index values calculated. Even changes in index values of 50 appeared as indistinguishable using our map projections (see Supplementary Materials, Figure S3), whereas changes of this magnitude are highly important for deterioration [45,53]. Therefore, Figure 8 shows the Scheffer Index for the period 1850–1879 (Figure 8a) and how it is projected to change from this historic baseline by the end of the 21st century (Figure 8b).

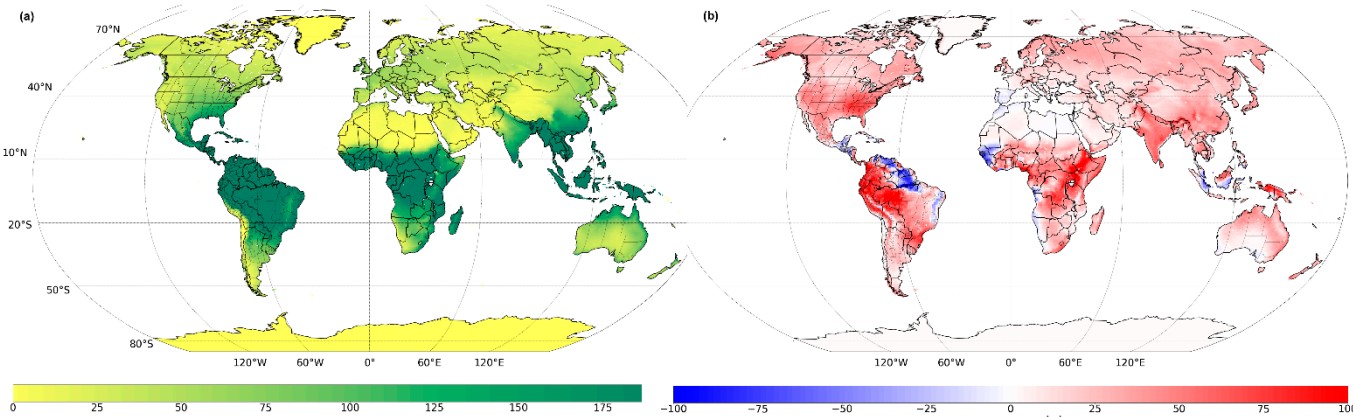

**Figure 8.** Scheffer Index (*Sch*) (**a**) in 1850–1879 and (**b**) the projected change in Scheffer Index between the historic past (1859–1879) and the far future (2070–2099). See Supplementary Materials, Figure S3 for maps of the four time periods. See also Video S7 (as an animated GIF) in Supplementary Materials.

Figure 8a shows that the greatest risk associated with the Scheffer Index is located in the region 20 degrees north and south of the Equator, with areas having an index score of more than 175. Minimal risk is found in polar regions and hyper arid drylands. Large areas of Europe, Central and Eastern Asia, Australia and northern North America have a Scheffer Index above 50. In previous studies, the value for high decay risk has varied with suggested critical index values including 48 [53] and 65 [45]. As such, even areas within our map that have comparably moderate index values still face notable risks.

By the end of the 21st century, projections show that the spatial area with an index value of more than 175 will continue to increase on all continents (e.g., along the Himalayas, northwards into Central China, south-eastern USA and the northwest coast of Australia). These changes affect a large range of timber heritage including large temples and pagodas in China to plantation houses in southern US states. Smaller but still notable increases in the Scheffer Index are projected across Europe where the index has increased from 50 (1850–1879) to 75 (2070–2099), which means according to the thresholds previous outlined by Lisø et al. [45] and Tajet and Hygen [53] this region is projected to fall within the high-risk threshold by the end of the century.

Changes in temperature have previously been identified as the primary driver of the Scheffer Index in Japan [29]. In areas between 0 and 60° N/S with large increases in the Scheffer Index (>+30), we similarly find temperature as the main driver of change over the 21st century (Figure 9a,b). However, for areas where there have been smaller increases,

or reductions in the Scheffer Index, changes in rainfall will likely be the primary driver. Polewards of 60°, large increases in the Scheffer Index ($\leq 100$) are projected to be caused by increases in rainfall (Figure 9c).

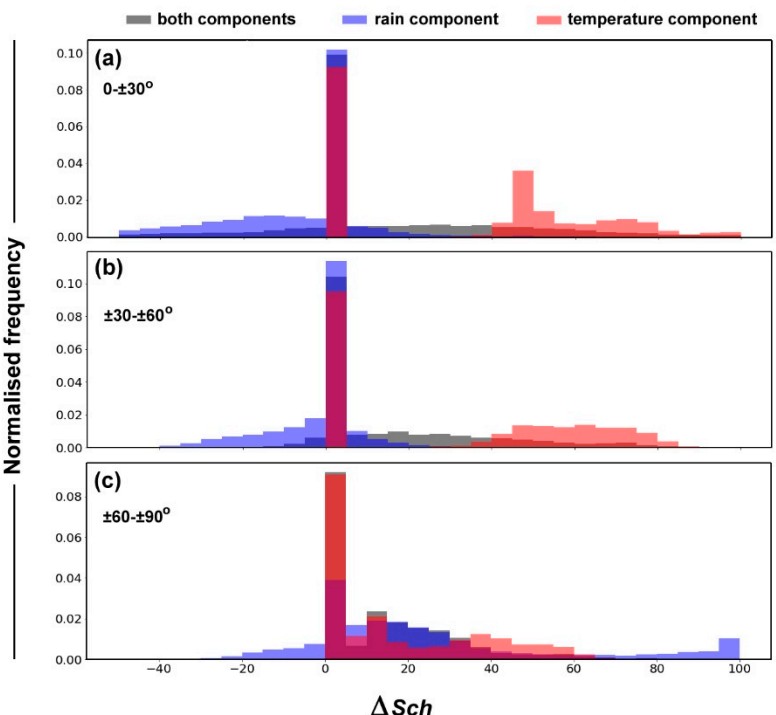

**Figure 9.** The normalised frequency of Scheffer Index values ($\Delta Sch$) between the far future (2070–2099) and the near past (1984–2013) when assessing (i) all components of the Scheffer Index (grey), (ii) the rain component (blue) and (iii) the temperature component (red) for areas at latitudes (**a**) 0–±30°, (**b**) ±30–±60° and (**c**) ±60–±90°.

## 4. Discussion

Previous studies of sites and regions broadly agree with our results. For example, the UK is likely to experience periods that are less humid than at present, in alignment with earlier findings, which showed that the UK would experience a $\leq 10\%$ decrease in RH and a reduction in ToW [54,55]. If the drying of timber is prolonged, it could reduce the risk posed by rot that requires damp material, and insect infestation may be limited if wood is very dry [12]. An increase in the RH range was generally caused by less humid conditions in the drier months, increasing physical stress. Additionally, global shifts in the Scheffer Index over the next century align with previous country-wide assessments [7,29,46,55].

Our study presents a picture of a less humid heritage climate that contrasts with a frequent view of a future wetter world [32]. There is a need to distinguish between specific humidity and relative humidity. The amount of water in the atmosphere (specific humidity) may increase under a warmer climate and hence drive heavier precipitation. However, higher temperatures mean the atmosphere can hold more moisture in a given volume of air, so that relative humidity could nevertheless be lower. Importantly, it is relative humidity that is a critical parameter for the water content of wood, fungal growth and insect activity [1]. We need to consider both temperature and moisture when assessing RH pressures, as changes to these can operate in the same or opposite directions, amplifying or suppressing the impact on timber deterioration.

### 4.1. Spatial and Temporal Scales

Understanding the global threat of moisture to historic timber enables a strategic approach for implementing effective conservation and management. Previous work assessing the environmental and climatic pressures on timber heritage has been undertaken

predominantly for specific sites [6,28] or at regional scales [45,46]. These smaller scales are useful for capturing local variations in climate that will affect specific sites and objects and particularly local issues pertaining to their use, such as the need for heating [56]. Recent findings of Brischke and Selter [6] highlight the importance of small-scale mapping at high resolution to account for decay processes in wood, due to variations imposed by local topographical and hydrological conditions. Finer spatial scales can limit the comparison between geographically, politically or economically disparate locations. Indeed, by using a global scale assessment, our findings show that many disparate areas can face similar threats to timber heritage. The identification of shared threats provides an exciting opportunity for new collaborations in strategic research and management, even where these do not arise from contiguous regions. This approach therefore speaks to the need raised by Richards and Brimblecombe [57] that if models are to be useful within heritage science, the model or model results need to be transferable into practice or aid policy decision making. In a subsequent paper, we explore changes in climate at specific sites, with particular types of coatings, wood and buildings [39], enabling us to further assess the tensions between the fine details provided by site scale approaches with the need for a strategic understanding of processes.

In addition, a global approach can show the pressures facing regions that have been less intensively studied to date. Our approach speaks to the needs highlighted by Orr et al. [37] and Simpson et al. [38] that advocate for additional research to address the imbalance in the geographical distribution of heritage research. Our results both broaden our understanding of risks facing timber heritage and highlight regions that may require additional research focus and identify narrow regions that are sensitive to change, which could easily be overlooked if a global scale approach was not taken.

### 4.2. Dominant Pressure

Figures 1, 3–5 and 7 illustrate the multiplicity of drivers of timber deterioration. It is difficult to assess combined or synergistic pressures without considering the specific nature of the interactions between climate and the wooden elements at sites, many of which are unknown. Instead, we identify the dominant pressure present in a given area. We define this dominance in a specific area, relative to the global distribution (as defined in Figure 10). However, the dominant pressure does not directly transfer into a quantifiable deterioration risk, as it does not account for thresholds in damage mechanisms. Nevertheless, it gives the geographical distribution of key processes that are likely to be important for conservation.

Figure 10 shows changes in the dominant pressure on heritage across the globe. In the future, there will be increases in the area dominated by fungal risk, most notably the Amazon and Congo Basins. However, for many regions, the RH range remains the dominant pressure across the period 1850–2099. Wind-driven rain is constrained to localised regions (e.g., Eastern Canada). Such enhanced ranges may place well-ventilated wooden interiors at risk, but these also represent locations that will become drier in the future. This figure also highlights the need for management strategies to recognise that pressures on historic timber may shift from physical to biological threats. As an example, the dominant pressure in Central Africa and Southeast Asia shifts from a historic threat from ToW to fungal risk by the end of the 21st century. This increasing importance of the Sheffer Index could suggest that issues relating to mould and insects may become more prevalent by the end of the 21st century, thus requiring new management strategies.

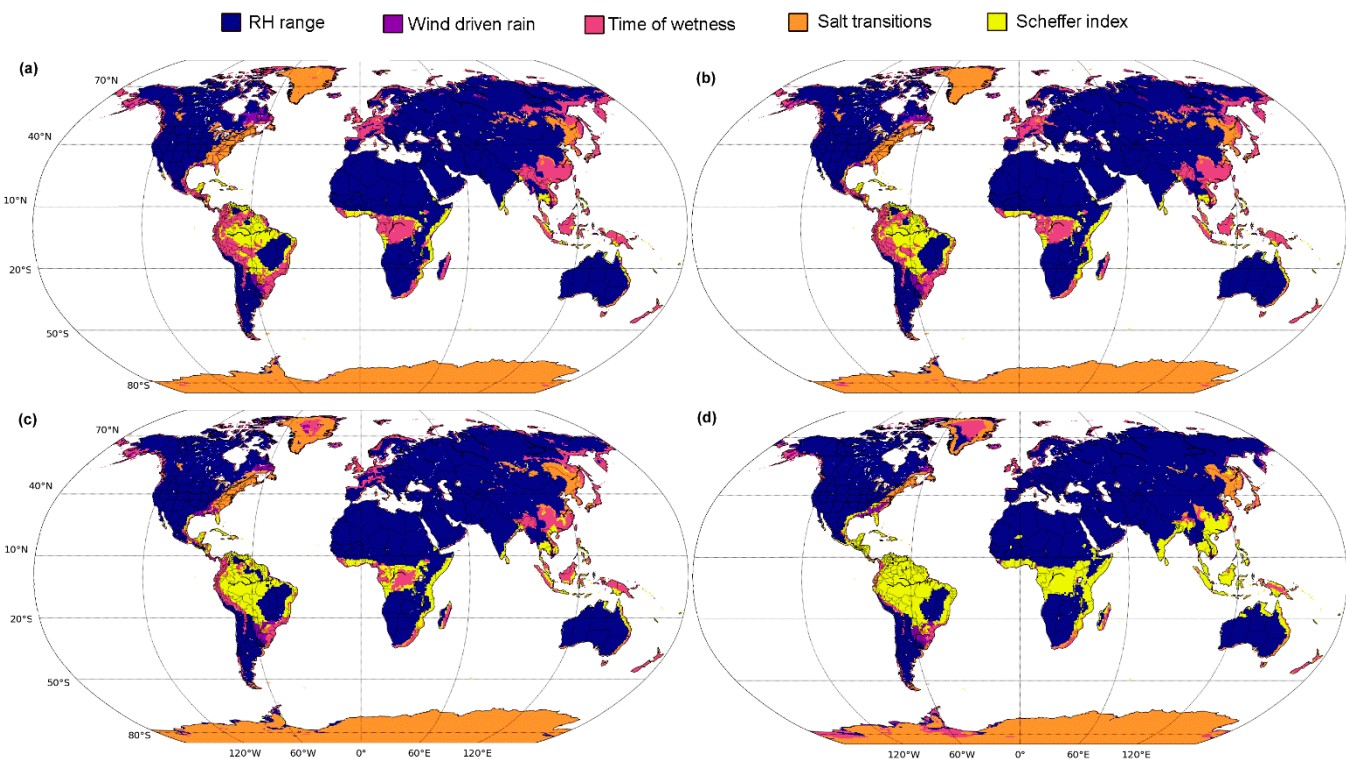

**Figure 10.** Dominant heritage climate pressure on timber (**a**) 1850–1879, (**b**) 1984–2013, (**c**) 2025–2054 and (**d**) 2070–2099. The dominant pressure was determined by comparing the percentile value for each of five pressures (excludes RH seasonality) for a given location compared to the global terrestrial datasets. The pressure with the greatest percentile value was assigned as the dominant pressure. For example, if in a given location the pressure percentiles were: RH range = 91%, WDR = 72%, ToW = 10%, Salt = 25% and Scheffer = 85%, the RH range would be assigned as the dominant pressure. See also Video S8 (as an animated GIF) in Supplementary Materials.

### 4.3. Propagation of Errors and Implication for Management

Our study engages with models to provide both the input data (HadGEM3-GC31-MM) and quantify climate pressures on timber heritage. The outcomes of any heritage model are not a perfect representation of reality [57], and this inevitably means that there will be differences between the modelled pressures and that experienced at specific heritage sites [39].

Climate models are less effective at capturing the climate in some regions; e.g., there is a lack of consensus over the sign of directional change in modelled precipitation and biases in surface wind speed over Central and West Africa [58,59]. While developments in regional climate modelling will improve the resolution of localised processes and the accuracy of model outputs, it is likely that present projections differ depending on the model selected [58]. Some climate parameters used in our assessment, such as temperature and humidity, are less affected by specific weather tracks, so they may be more resistant to error [60]. However, we need to be mindful that the results from regions associated with model uncertainty may be sensitive to model choice, so future use of an ensemble of models is likely to reduce the uncertainty in projected heritage climates.

Our study, which focussed on a future worst-case scenario, aimed to provide an understanding of the nature of the threat to timber. Even if the worst-case is never realised, this approach gives a sense of what threats likely need to be managed. Additionally, it also highlights regions that might be sensitive to tipping points or thresholds. Identifying regions that are projected to undergo large changes, allows resources to be appropriately distributed in addressing threats in a timely fashion. Change in climate pressures across

the seasons may be important in heritage management, as cultural events, visitor activities and the landscape are strongly dependent on time of year [49].

*4.4. Strategic Policy for Timber Heritage*

Our approach has highlighted the challenge of developing meaningful heritage climate parameters at an appropriate scale for policy or management. We assessed the pressures at a global scale, enabling inter-regional comparisons to be made. This approach fills the need for a strategic approach to heritage climate pressures. However, it is hard to interpret global projections in terms of managing the threat posed to individual sites, buildings or objects, so these are explored in Brimblecombe and Richards [39].

As heritage is constantly interacting with and being influenced by environmental surroundings [61], heritage policymakers, managers and practitioners must constantly plan and react to the inevitability of change at heritage sites. Being able to identify regions that are likely to experience future change or cross thresholds for decay processes can enable conservation strategies to be preventative, rather than reactive. For timber heritage, regions likely to experience notable increases in moisture-related pressures, this global approach could allow international collaboration between regions experiencing similar climatic decay processes.

Our study shows the benefits of including heritage climate parameters within policy. Traditional parameters used in climate science, such as average change in temperature or specific humidity over a 30-year period, can miss key interactions between heritage and climate that drive deterioration. If further developed, the notion of heritage climatology could extend to the concepts of risk that are more directly transferable to heritage practice. By adopting climate parameters attuned to heritage, institutions, such as the UNESCO WHC, could be in a better position to embrace the notion of climate change impacts on heritage.

## 5. Conclusions

We conclude that the future heritage climate will lead to a redistribution of risks to timber heritage under a high emission scenario. The changes in the magnitude and spatial distribution of the pressures in this heritage reflect shifts in the relative humidity range, seasonality, time of wetness, wind-driven rain, salt transitions and potential for fungal attack. It is noticeable that in the future, tropical areas, such as the Congo and Amazon Basins, along with parts of Southeast Asia show less humid conditions. Some temperate regions will show increasing humidity ranges, with the most humid months shifting earlier in the year. We find projected increases in days of wind-driven rain for some temperate regions. Salt transitions may become more evident in drylands. By 2070, a broad extension of fungal attack on timber will spread from the Himalayas into Central China, driven as much by temperature as rainfall. Our picture presents a less humid global heritage climate, and it contrasts with common views that the future will be wetter. However, it is important to distinguish specific and relative humidity as the former is likely to increase with temperature, while the latter is likely to decrease. Relative humidity is of particular importance to timber heritage because this parameter affects fungal growth, insect activity and water content of timber.

Our study used a single climate model under a worst-case scenario, giving us a sense of the direction of change. However, future work might address uncertainties associated with climate models by using ensemble outputs. Additional work could also focus on how multiple pressures synergistically interact to increase timber deterioration, particularly in regions where the dominant process is projected to shift over the coming century. More collaboration between heritage researchers and practitioners might help clarify the incorporation of uncertainties associated with projections into management plans.

Our model outputs used global maps, which can be a useful tool for dissemination and discussion both in terms of (i) policy making, particularly at a strategic level and (ii) raising public awareness by providing visual representations. Shifts in threats over

time can provide an engaging narrative for a public interested in climate change. However, heritage is frequently managed at the level of a specific site, so such broad pictures will hopefully provoke thoughts about choices to be made at the site level.

**Supplementary Materials:** The following supporting information can be downloaded at: https://www.mdpi.com/article/10.3390/heritage5030100/s1: Figure S1: Modal month with the maximum monthly RH for the periods (a) 1850–1879, (b) 1984–2013, (c) 2025–2054 and (d) 2070–2099; Figure S2: Modal month with the minimum monthly RH for the periods (a) 1850–1879, (b) 1984–2013, (c) 2025–2054 and (d) 2070–2099; Figure S3: Scheffer Index for the periods (a) 1850–1879, (b) 1984–2013, (c) 2025–2054 and (d) 2070–2099; Video S1: Animated GIF of Figure 1 showing RH range (a) 1850–1879, (b) 1984–2013, (c) 2025–2054 and (d) 2070–2099; Video S2: Animated GIF of Figure 3 showing modal month with the maximum monthly RH for the periods (a) 1850–1879, (b) 1984–2013, (c) 2025–2054 and (d) 2070–2099. Video S3: Animated GIF showing modal month with the minimum monthly RH for the periods (a) 1850–1879, (b) 1984–2013, (c) 2025–2054 and (d) 2070–2099. Video S4: Animated GIF of Figure 4 showing ToW for the periods (a) 1850–1879, (b) 1984–2013, (c) 2025–2054 and (d) 2070–2099. Video S5: Animated GIF of Figure 5 showing WDR for the periods (a) 1850–1879, (b) 1984–2013, (c) 2025–2054 and (d) 2070–2099. Video S6: Animated GIF of Figure 7 showing salt transitions for the periods (a) 1850–1879, (b) 1984–2013, (c) 2025–2054 and (d) 2070–2099. Video S7: Animated GIF of Figure 8 showing the Scheffer Index for the periods (a) 1850–1879, (b) 1984–2013, (c) 2025–2054 and (d) 2070–2099. Video S8: Animated GIF of Figure 10 showing combined pressures for the periods (a) 1850–1879, (b) 1984–2013, (c) 2025–2054 and (d) 2070–2099.

**Author Contributions:** Conceptualization, formal analysis, investigation and writing—review and editing, P.B. and J.R.; software and data curation, J.R.; original draft preparation P.B. All authors have read and agreed to the published version of the manuscript.

**Funding:** This research received no external funding.

**Institutional Review Board Statement:** Not applicable.

**Informed Consent Statement:** Not applicable.

**Data Availability Statement:** Data are available from the links noted in the text.

**Acknowledgments:** We would like to thank Sebastian Engelstaedter for his technical support with the climate model data.

**Conflicts of Interest:** The authors declare no conflict of interest.

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
