# Peer review of "Moisture as a Driver of Long-Term Threats to Timber Heritage—Part I: Changing Heritage Climatology"

_heritage, doi:10.3390/heritage5030100_

Round 1
Reviewer 1 Report
This is a good and relevant paper although some changes are suggested.
General comment. It would be easier to read the paper if the scope is better defined. What do you mean by timber heritage? Wooden-built heritage?
Paragraph:
1.2 '....crack, posing threats to both the structural integrity of a timber building or object and the aesthetic value of any detailing'. I don't think this statement is true or necessary. The outdoor timber structures are heavily cracked forming a natural ‘patina’ which is an intrinsic feature of the wooden heritage objects, so the impact on aesthetical value is questionable. It is known in the engineering field that cracks do not change significantly the usability of the timer as a building material. I would remove this sentence as it is stating something which is not valid for the majority of wooden heritage objects stored outdoors.
‘Salts can also arise
from groundwater or due to the activities occurring within the buildings (e.g. fish curing)’ – the main reason for the salt presence in buildings located inland is the urine of domesticated animals. Therefore, nitrates are much more frequently identified salt in buildings.
‘In addition, we do not focus on outdoor timber, so do not discuss indoor heritage [9,30]’ – the sentence is unclear. What is the focus then?
The authors could add the reference to the paper https://doi.org/10.1007/s00107-022-01841-3 which indicates that due to the nonlinear dependence of the moisture diffusion coefficient on RH as well as moisture sorption the change in RH variation will generate the change in moisture content of the wood. Although, I don’t know how relevant is the effect.
Paragraph 2
The introduction gives a promise that the risk of insect-driven deterioration will be analyzed but sadly there is nothing in the paper except a few qualitative sentences. The authors stated the dependence of insect activity on RH but there is no RH (or RH-related relevant parameter) map helping to understand how and where the risk will change. If it is outside the scope of the paper please state it clearly at the beginning.
Paragraph 3. The maps presenting the data are very difficult to work with, particularly if one wants to see the change in the parameter of interest. I would suggest adding the maps showing the difference between presence and future as you did in figure 8.
Fig. 9 the description should be ‘The probability….’
Fig. 10 As I have stated above, I don’t think RH variations are the relevant parameter for decision makers (can you give one example of a preventive conservation measure undertaken to reduce the cracking of timber structures? Or can you give an example of the wooden timber structure damaged by RH variations that is supported by sound evidence?) Therefore, I would add also a second the most important risk as RH variation dominates a significant part of the map.
Author Response
REFEREE 1
This is a good and relevant paper although some changes are suggested.
General comment. It would be easier to read the paper if the scope is better defined. What do you mean by timber heritage? Wooden-built heritage?
AGREE so the title has been changed "Moisture as a driver of long-term threats to timber heritage" AND Section 1.3. Approach now starte "This study investigates the moisture related pressures imposed by climate on timber heritage, here taken primarily as outdoor buildings."
Paragraph:
1.2 '....crack, posing threats to both the structural integrity of a timber building or object and the aesthetic value of any detailing'. I don't think this statement is true or necessary. The outdoor timber structures are heavily cracked forming a natural ‘patina’ which is an intrinsic feature of the wooden heritage objects, so the impact on aesthetical value is questionable. It is known in the engineering field that cracks do not change significantly the usability of the timer as a building material. I would remove this sentence as it is stating something which is not valid for the majority of wooden heritage objects stored outdoors.
AGREE so changed to "Persistent variations in relative humidity and moisture interacting on the carved wood surfaces (e.g. in sculpture) can cause such objects to weaken and crack"
‘Salts can also arise
from groundwater or due to the activities occurring within the buildings (e.g. fish curing)’ – the main reason for the salt presence in buildings located inland is the urine of domesticated animals. Therefore, nitrates are much more frequently identified salt in buildings.
AGREE we have added a note about this, but the other salts form part of some work on the thermodynamics of salts on wood with the University of Ghent, to be written up in September. The sentence now reads "One of the most common salts is sodium chloride, which dissolves or crystallises at 75.5% relative humidity and derived from sea spray and road salts [], but in buildings located inland nitrates and urea contribute to salts present . "
‘In addition, we do not focus on outdoor timber, so do not discuss indoor heritage [9,30]’ – the sentence is unclear. What is the focus then?
THIS WAS A MISTAKE, THANKS FOR SPOTTING IT!
The authors could add the reference to the paper https://doi.org/10.1007/s00107-022-01841-3 which indicates that due to the nonlinear dependence of the moisture diffusion coefficient on RH as well as moisture sorption the change in RH variation will generate the change in moisture content of the wood. Although, I don’t know how relevant is the effect.
AGREE -- WE HAD NOT SEEN "Soboń M, Bratasz Ł. A method for risk of fracture analysis in massive wooden cultural heritage objects due to dynamic environmental variations. European Journal of Wood and Wood Products. 2022 Jun 19:1-3." BUT NOW ADDED. WE KNOW LUKAS WELL.
Paragraph 2
The introduction gives a promise that the risk of insect-driven deterioration will be analyzed but sadly there is nothing in the paper except a few qualitative sentences. The authors stated the dependence of insect activity on RH but there is no RH (or RH-related relevant parameter) map helping to understand how and where the risk will change. If it is outside the scope of the paper please state it clearly at the beginning.
AGREE - GOOD POINT, THOUGH AT TIMES WE TALKED ABOUT TERMITES. WE HAVE NOW TIED TO TIME OF WETNESS AND NOW EMPHASISED E.G.
"laboratory evaluation of insect (e.g. termites) damage and consumption to wood [42]."
"the Congo, Kisangani (formerly Stanleyville), the capital of Tshopo province, where timber has been used in both vernacular and old colonial buildings. Such decreases in time of wetness could reduce the risk of insect damage as timber will have shorter periods of high moisture content"
"This continued pressure from lengthy periods of wetness means that the risk of insect attack "
Paragraph 3. The maps presenting the data are very difficult to work with, particularly if one wants to see the change in the parameter of interest. I would suggest adding the maps showing the difference between presence and future as you did in figure 8.
WHEN PREPARING THE PAPER THIS TROUBLED US ENORMOUSLY SO WE CREATED ANIMATED GIFS AS MOVIES IN THE SUPPLEMENT. CHANGES BETWEEN PANES IN THE FIGURES ARE SMALL, BUT WE NOW MENTION THIS IN THE TEXT. "Changes can be subtle, especially in the past, so we provide animated versions of the global maps in the supplement " THE ANIMATIONS HAVE NOW BEEN DONE FOR ALL FIGURES AND NOTED IN EACH FIGURE CAPTION. THE ONLINE VERSION HAS TIFF FILES THAT CAN BE ENGLARGED AS REQUIRED AND ARE AT HIGH RESOUTION. WE ARE ALWAYS RELUCTANT TO OVERUSE DIFFERENCES AS THESE CAN AMPLIFY CHANGES THAT ARE NOT SIGNIFICANT BECAUSE THE ABSOLUTE RISK/PRESSURE MAY BE SMALL.
Fig. 9 the description should be ‘The probability….’
GOOD POINT Figure 9. The normalised frequency of difference in Scheffer Index values CHANGED
Fig. 10 As I have stated above, I don’t think RH variations are the relevant parameter for decision makers (can you give one example of a preventive conservation measure undertaken to reduce the cracking of timber structures? Or can you give an example of the wooden timber structure damaged by RH variations that is supported by sound evidence?) Therefore, I would add also a second the most important risk as RH variation dominates a significant part of the map.
AGREE SO WE HAVE NOW EMPHASISESD THAT THE RANGE LARGELY REFLECTS A DRYING, SO THE BLUE AREAS OF THE MAPP AS AS MUCH ABOUT THIS. DAMAGE IS COMMO TO SMALLER OBJECTS SUCH AS SCULPTURE - SO INDOOR HUMIDITY RANGE IS LMITED SO ADD "Such enhanced ranges may place well-ventilated wooden interiors at risk, but these also represent locations which will become drier in the future. "
Reviewer 2 Report
“Water and moisture relations are especially important for wood. These can drive
physical changes, and mediate biological and chemical processes that cause deterioration
in timber [2].”
Comment: There are many good references to support this statement. Why was this 2015 one chosen? Is this the only reference you have read on this subject? To understand the effect of moisture on wood properties, you need to expand your knowledge.
“Rot is another common process threatening timber heritage. Fungal attack is
mediated by climate factors: temperature, water or exposure to high humidity [18].”
Comment: As before, there are many good references on this subject before 2021. Is this the only reference you have read?
Comment: There is a very poor literature review on the deterioration of wood in a moist environment. The whole point of this manuscript is to avoid moisture but the authors do not have a good understanding of the effects of moisture on wood.
“a scenario based on a high emission future.”
Comment: This assumes that there will be no change in global emission standards?
Question: What percentage of structures have some sort of protection build in: preservative, paint, etc. This would change the entire analysis.
Comment: I can not see any differences in the small images shown in Figure 1, 4, 5, 7?
“time periods (a) 1850-1879, (b) 1984-2013, (c) 2025-2054 and (d) 2070-2099]”
Comment: There is good data for (a) and (b) and a projection into (c) but (d) is a long time in the future and the climate might change so I wonder if this time period is relevant?
There a many unanswered questions about climate change in the future. While I find this data interesting, I am not sure a report, this detailed, on an unknown future should occupy this much page space.
Author Response
REFEREE 2
“Water and moisture relations are especially important for wood. These can drive physical changes, and mediate biological and chemical processes that cause deterioration in timber [2].”
Comment: There are many good references to support this statement. Why was this 2015 one chosen? Is this the only reference you have read on this subject? To understand the effect of moisture on wood properties, you need to expand your knowledge.
“Rot is another common process threatening timber heritage. Fungal attack is mediated by climate factors: temperature, water or exposure to high humidity [18].”
Comment: As before, there are many good references on this subject before 2021. Is this the only reference you have read?
Comment: There is a very poor literature review on the deterioration of wood in a moist environment. The whole point of this manuscript is to avoid moisture but the authors do not have a good understanding of the effects of moisture on wood.
WE AGREE THAT THE REFERENCES HERE WERE POORLY CHOSEN, SO WE HAVE PRMOTED THE TEXTBOOK (56. Skaar C. Wood-water relations. Springer Science & Business Media 2012 ) TO THE BEGINNING.
“a scenario based on a high emission future.”Comment: This assumes that there will be no change in global emission standards?
GOOD POINT SO WE HAVE EMPHASISE THIS IS NOT SO MUCH A REAL FUTURE, BUT A WORST-CASE SCENARIO AS NOW EMPHASISED e.g. "emission future (i.e. to reflect a worst-case scenario). "
Question: What percentage of structures have some sort of protection build in: preservative, paint, etc. This would change the entire analysis.
AGREE PAINT IS VERY MUCH PART OF THE SECOND MS ON THIS TOPIC REFERENCED AS: "39. Brimblecombe P & Richards J., Moisture as a driver of long-term threats to timber heritage: Moisture as a driver of long term threats to historic timber: II risks imposed at local sites, Heritage " NOTED AS "No distinction is made between untreated timber and that with pesticides or surface coatings, although these would respond more slowly to climate and biological threats, but treated in more detail in [39].
Comment: I can not see any differences in the small images shown in Figure 1, 4, 5, 7?
THIS HAS TROUBLED US ENORMOUSLY SO WE CREATED ANIMATED GIFS AS MOVIES IN THE SUPPLEMENT. CHANGES BETWEEN PANES IN THE FIGURES ARE SMALL, BUT WE NOW MENTION THIS IN THE TEXT. "Changes can be subtle, especially in the past, so we provide animated versions of the global maps in the supplement " THE ANIMATIONS HAVE NOW BEEN DONE FOR ALL FIGURES AND NOTED IN EACH FIGURE CAPTION. THE ONLINE VERSION HAS TIFF FILES THAT CAN BE ENGLARGED AS REQUIRED AND ARE AT HIGH RESOUTION. WE ARE ALWAYS RELUCTANT TO OVERUSE DIFFERENCES AS THESE CAN AMPLIFY CHANGES THAT ARE NOT SIGNIFICANT BECAUSE THE ABSOLUTE RISK/PRESSURE MAY BE SMALL.
“time periods (a) 1850-1879, (b) 1984-2013, (c) 2025-2054 and (d) 2070-2099]”
Comment: There is good data for (a) and (b) and a projection into (c) but (d) is a long time in the future and the climate might change so I wonder if this time period is relevant?
THE REASONING BEHIND THE FOUR PERIODS IS NOW EXPLAINED. " four 30-year time periods 1850-1879 (historic- to set a baseline); 1984-2013 (recent past- available data sets); 2025-2054 (near future- long-term planning horizon) and 2070-2099 (far future- sense of overall direction)" THESE ARE OFTEN ADOPTED IN CLIMATE CHANNGE PUBLICATIONS AND FOLLOW THAT OF HERITAGE RELATED PROJECTS SUCH AS NOAH'S ARK AND CLIMATE4CULURE (9. Leissner, J.; Kilian, R.; Kotova, L.; Jacob. D.; Mikolajewicz, U.; Broström, T.; Ashley-Smith, J.; Schellen, H.L.; Martens, M.; van Schijndel, J.; Antretter, F.; Climate for Culture: assessing the impact of climate change on the future indoor climate in historic buildings using simulations. Herit. Sci. 2015 , 3, 1-5. 3. Sabbioni, C.; Brimblecombe, P,; Cassar, M. editors. The atlas of climate change impact on European cultural heritage: scientific analysis and management strategies. Anthem Press, London; 2010.)
There a many unanswered questions about climate change in the future. While I find this data interesting, I am not sure a report, this detailed, on an unknown future should occupy this much page space.
THE FUTURE IS UNKNOWABLE, BUT PROJECTIONS OF THIS KIND DOMINATE THE IPCC (Intergovernmental Panel on Climate Change) REPORTS, SO WE BELIEVE THEM TO BE REASONABLY RELIABLE. THE ISSUE OF ERROR IS EXPLICITLY ADDRESSED IN "4.3. Propagation of errors and implication for management". ON THE MORE GENERAL ISSUE OF ERRORS IN HERITAGE CLIMATE PROJECTIONS, WE ARE CURRENTLY RUNNING ENSEMBLE MODELS - THESE ARE RELEVANT FOR CLIMATES SUCH AS THOSE IN AFRICA WHICH ARE MORE PRONE TO ERROR: "Climate models are less effective at capturing the climate in some regions e.g. there is a lack of consensus over the sign of directional change in modelled precipitation and biases in surface wind speed over central and west Africa [58,59]." AND AS WE NOTED THIS IN THE CONCLUSIONS ". However, future work might address uncertainties associated with climate models, by using ensemble outputs. "
Round 2
Reviewer 2 Report
I still can not see visual differences in the small figures. I suggest you show one figure and describe changes using that figure, i.e.. what is expected to change in what area during what time. This would make it much easier to understand and save page space.
Author Response
Yes we can understand the need for changing climate pressure to be clear to the reader, so each map needs to be introduced. This has be achieved for Figs. 1, 2, 4, 5, 8 and 10, by modifying the introductory paragraph to each section such that it emphasises the global distribution of the change to be seen on the maps.